Proceedings of the 7th Symposium on Advances in Approximate Bayesian Inference, 2025 1–26

# Divide, Conquer, Combine Bayesian Decision Tree Sampling

**Jodie A. Cochrane**  Jodie.Cochrane@newcastle.edu.au
**Adrian Wills**  Adrian.Wills@newcastle.edu.au
**Sarah J. Johnson**  Sarah.Johnson@newcastle.edu.au
*University of Newcastle*

## Abstract

Decision trees are commonly used predictive models due to their flexibility and inter-pretability. This paper is directed at quantifying the uncertainty of decision tree predictions by employing a Bayesian inference approach. This is challenging because these approaches need to explore both the tree structure space and the space of decision parameters associated with each tree structure. Importantly, the structure and the decision parameters are tightly coupled; small changes in the tree structure can demand vastly different decision parameters to provide accurate predictions. A challenge for existing sample-based approaches is proposing joint changes in both the tree structure and the decision parameters that result in efficient sampling. This paper takes a different approach, where each distinct tree structure is associated with a unique set of decision parameters. The proposed approach, entitled DCC-Tree, is inspired by the work in Zhou et al. (2020) for probabilistic programs and Cochrane et al. (2023) for Hamiltonian Monte Carlo (HMC) based sampling for decision trees. Results show that DCC-Tree performs comparably to other HMC-based methods and better than existing Bayesian tree methods while improving on consistency and reducing the per-proposal complexity.

## 1. Introduction

The decision tree model is used extensively in various industries as it provides both flexibility to describe data relations and model interpretability. Decision trees define a set of hierarchical splits that partition the input space into a union of disjoint subspaces. Datapoints associated with each subspace are assumed to originate from the same distribution, defining the model predictions. Standard decision tree methods are CART (Breiman et al., 1984), ID3 (Quinlan, 1986), and C4.5 (Quinlan, 1993), which provide point-estimate predictions.

It is important for any decision-making process to consider the uncertainty associated with a point-estimate prediction. As such, there has recently been growing emphasis on estimating the prediction uncertainty for machine learning models (Ghahramani, 2015). Under the Bayesian paradigm, probability distributions are used to define anything about the model that is not known for certain, including the model parameters themselves. The uncertainty in the model parameters induces uncertainty in the resulting predictions, which can be used to estimate confidence intervals via evaluation of an expectation integral.

However, a current challenge in applying Bayesian inference to decision trees is that a change in the tree structure fundamentally changes the meaning of the parameter vector. This issue manifests in the coupling between the tree topology and the associated decision parameters. Evaluating the expectation integral therefore requires consideration of both the nonlinearity of the model and the association between the parameter vector and the

tree topology. A common approach is to approximate the integral based on useful samples from the parameter distribution using a method known as Monte Carlo integration. The problem then shifts to how to generate these useful samples for the decision tree model.

To date, the literature has focused on Markov chain Monte Carlo (MCMC) approaches to address this problem and attempt to explore the posterior distribution of the decision tree parameters. Random-walk MCMC was first proposed by Chipman et al. (1998) and Denison et al. (1998) where a set of local, tree-inspired proposals are used to move around the space. Over time, improvements have been made to that original set of proposal methods, but have remained based on random-walk methods (Wu et al., 2007; Gramacy and Lee, 2008; Pratola et al., 2016). Separately, there have been attempts to use Sequential Monte Carlo (SMC) sampling methods to explore the posterior distribution (Taddy et al., 2011; Lakshminarayanan et al., 2013). However, these transitions are similar to the localised, random-walk proposals of MCMC-based methods, although Lakshminarayanan et al. (2013) does provide a locally optimal proposal option to try and improve on the random-walk issues.

The recent work of Cochrane et al. (2023) has shown the efficacy of using Hamiltonian Monte Carlo (HMC) within the MCMC framework to explore the decision tree space, where the decision tree structure has been softened to enable full use of the HMC benefits while remaining interpretable. HMC is a more efficient sampling method that uses gradient information from the likelihood to generate subsequent samples (Neal et al., 2011; Betancourt, 2017). Although the benefits of using HMC are clear, the authors note that one main impediment to their method is the number of intermediate HMC samples required at every iteration, drastically increasing computational overhead for each proposal.

In this paper, we investigate a different perspective by which to explore the posterior distribution of decision trees. Inspired by the work of Zhou et al. (2020), which proposes the Divide, Conquer, Combine (DCC) inference framework in the context of probabilistic single-line programs, the parameter space of the decision tree is described by an all-encompassing parameter vector with constant dimension. This is in contrast to previous Bayesian decision tree algorithms, where the dimension of the parameter vector varied throughout the sampling routine. Parameters corresponding to all possible tree topologies are included in the parameter vector, with a subset of this vector considered at each iteration, such that each distinct tree structure is associated with a unique set of decision parameters. This concept forms the basis of the novel DCC-Tree sampling method presented in this paper.

## 2. Preliminaries

We consider the situation in which the model is assumed to be a decision tree and we wish to determine the uncertainty on the decision tree parameters. Although relatively easy to use to make point-wise predictions, defining uncertainty for the predictions from a decision tree model is a more difficult task. This section will first define the decision tree model and then an alternative parameterisation required for the implemented sampling method.

### 2.1. Decision Trees

A standard binary decision tree is parameterised as

$$\mathbb{T} = (\mathcal{T}, \boldsymbol{\kappa}, \boldsymbol{\tau}, \boldsymbol{\theta}), \tag{1}$$

where: $\mathcal{T}$ denotes the tree topology, which encompasses information about the set of internal nodes $\eta_j \in \mathcal{I}, j = 1, \ldots, n$, the set of leaf nodes $\eta_{\ell_k} \in \mathcal{L}, k = 1, \ldots, n_\ell$, and the set of interconnections; $\boldsymbol{\kappa} = [\kappa_1, \kappa_2, \ldots, \kappa_n]$ denotes the splitting indices of each internal node $\eta_j \in \mathcal{I}$; $\boldsymbol{\tau} = [\tau_1, \tau_2, \ldots, \tau_n]$ represent the splitting thresholds for each internal node $\eta_j \in \mathcal{I}$; $\boldsymbol{\theta} = [\theta_1, \theta_2, \ldots, \theta_{n_\ell}]$ represents the leaf node parameters for $\eta_{\ell_k} \in \mathcal{L}$. We refer to the dataset as $\mathcal{D} = \{(\mathbf{x}_i, \mathbf{y}_i)\}_{i=1}^N$ which is comprised of $N$ sets of inputs $\mathbf{x}_i \in \mathcal{X}$ and outputs $\mathbf{y}_i \in \mathcal{Y}$. The input and output dimensionalities are denoted $n_x$ and $n_y$ respectively. In this paper, we will consider both regression and classification problems where $\mathcal{Y} = \mathbb{R}$ or $\mathbb{Z}$, and the input space is real-valued $\mathcal{X} = \mathbb{R}^{n_x}$.

The traditional approach to constructing a decision tree is via a greedy one-step-ahead heuristic. Predictions for a given input are made by traversing the tree, starting at the root node and recursively moving to one of the children nodes corresponding to the value of the relevant splitting indices for the given input. The traversal stops once a leaf node has been reached. The output is then related to the datapoints that are in that leaf node. In a standard decision tree, this corresponds to a point-estimate output prediction.

## 2.2. Trees as Disjoint Subspaces

The premise behind the DCC-Tree algorithm is to consider the sample space as a union of disjoint subspaces of varying dimensions. The method exploits this underlying structure to the problem in an attempt to explore the posterior distribution of decision tree parameters.

In this paper, we will use the variable $m$ to refer to a distinct tree topology that defines a specific subspace, i.e. $m \equiv \mathcal{T}_m$, with the remaining local parameters collected into the vector $\Theta_m$. This implies that the tree topology parameter is discrete (and can be considered categorical, i.e. no natural numerical/ordering). For the DCC-Tree method, the parameter vector is all-encompassing, that is,

$$\Theta = \begin{bmatrix} \Theta_1 & \Theta_2 & \ldots & \Theta_M \end{bmatrix}. \tag{2}$$

Here, $\Theta_m$ denotes the set of model parameters corresponding to the specific model structure $m$ which is indexed by the discrete random variable $\mathcal{M}$ with support $\{1, \ldots, M\}$. Note that this definition is in contrast to the parameter vector with varying dimensions commonly assumed in other Bayesian decision tree methods. The target distribution is the posterior on both the model parameters $\Theta$ and the discrete random variable $\mathcal{M}$.

If the discrete random variable takes on the value $m$, then the posterior distribution is proportional to the local parameter distribution (see Appendix A),

$$p(\Theta, m \mid \mathcal{D}) \propto p(\Theta_m \mid m, \mathcal{D}). \tag{3}$$

As a result, each tree topology can be considered separately with respect to the local inference method, and later combined appropriately to give an estimate of the overall space.

## 2.3. Soft Decision Tree Model Definition

One of the major difficulties in applying HMC to the decision tree model is the hard split parameterisation. To take full advantage of the HMC sampling method, we adopt the soft decision function and input selection (HMC-DFI) parameterisation used in Cochrane et al.

(2023), where both the hard split function and splitting indices ($\boldsymbol{\kappa}$) are softened. This section will summarise this information for the notation used in this paper.

Under the HMC-DFI specification, the splitting index $\kappa_j$ is instead parameterised as a unit simplex and denoted $\Delta_{\eta_j}$ for internal node $\eta_j$, where $\Delta_{\eta_j} = \left[ w_{\eta_j,1}, w_{\eta_j,2}, \ldots, w_{\eta_j,n_x} \right]$, with $w_{\eta_j,i} > 0$ and $\sum_{i=1}^{n_x} w_{\eta_j,i} = 1$. Therefore, if the discrete random variable takes on the value $m$, the corresponding local parameter vector is $\Theta_m = (\boldsymbol{\Delta}_m, \boldsymbol{\tau}_m, \boldsymbol{\theta}_m)$.

Further, the hard split function is softened such that each datapoint is no longer associated with a single leaf node, but instead now has an associated probability of being assigned to each leave node. For a specific tree subspace which we denote $m$, the probability that a datapoint $(\mathbf{x}_i, y_i)$ goes to the left at internal node $\eta_j$ is defined as,

$$\psi(\mathbf{x}_i \mid \Theta_m, \eta_j) = f\left( \frac{\mathbf{x}_i \Delta_{m,\eta_j} - \tau_{m,\eta_j}}{h} \right), \tag{4}$$

where $\Delta_{m,\eta_j}$ and $\tau_{m,\eta_j}$ denote the splitting index and splitting threshold respectively of the internal node $\eta_j$. Note that $f$ is any function that provides a soft approximation for binary splits (here the logistic function $f(x) = (1 + \exp(-x))^{-1}$), and $h$ is the corresponding split sharpness parameter (Cochrane et al., 2023). The probability that a datapoint is assigned to a specific leaf node can then be computed as the total probability along the path required to reach the leaf node. If we denote the probability of a datapoint $(\mathbf{x}_i, y_i)$ being assigned to leaf node $\eta_{\ell_k}$ in decision tree $m$ as $\phi_{i,m,k}$, the probability can be expressed as follows,

$$\phi_{i,m,k}(\mathbf{x}_i \mid \Theta_m, \eta_{\ell_k}) = \prod_{\eta \in \mathcal{A}(\eta_{\ell_k})} \psi(\mathbf{x}_i \mid \Theta_m, \eta)^{R_\eta} (1 - \psi(\mathbf{x}_i \mid \Theta_m, \eta))^{1-R_\eta}, \tag{5}$$

where $\mathcal{A}(\eta)$ and $R_\eta$ denote the set of ancestor nodes and direction vector for node $\eta$.

Prior Specification

The prior on the joint mixed discrete-continuous distribution can be split into a prior on the discrete variable $\mathcal{M}$ and a prior on the continuous parameters $\Theta_m$ for each subspace. The prior on the discrete variable is taken to align with the standard tree structure prior as originally defined in Chipman et al. (1998). This relates the probability of each tree subspace $m$ to the corresponding tree structure $\mathcal{T}_m$. Let $\mathcal{I}_m$ and $\mathcal{L}_m$ denote the set of internal and leaf nodes corresponding to the specific tree subspace $m$. The probability that the discrete random variable takes on the value $m$ is then defined as,

$$p(m) \propto \prod_{\eta_j \in \mathcal{I}_m} p_{\text{SPLIT}}(\eta_j) \times \prod_{\eta_{\ell_k} \in \mathcal{L}_m} (1 - p_{\text{SPLIT}}(\eta_{\ell_k})), \tag{6}$$

where $p_{\text{SPLIT}}(\eta)$ denotes the probability that a node $\eta$ will split. Again, the definition used in Chipman et al. (1998) is adopted here, where $p_{\text{SPLIT}} = \alpha(1 + d_\eta)^{-\beta}$, with hyperparameters $\alpha \in (0, 1)$ and $\beta \geq 0$ and where $d_\eta$ represents the depth of the node in the tree.

The priors on the local parameters for each tree subspace are defined independently based on the specific decision tree model under consideration. If the discrete random variable takes on the value $m$, then the prior can be expressed as

$$p(\Theta_m) = p_\Delta(\boldsymbol{\Delta}_m \mid m) p_\tau(\boldsymbol{\tau}_m \mid m) p_\theta(\boldsymbol{\theta}_m \mid m). \tag{7}$$

The discrete variable $m$ refers to a specific tree topology $\mathcal{T}_m$ which dictates the dimension of the remaining parameters (i.e. $\mathbf{\Delta}_m/\boldsymbol{\tau}_m/\boldsymbol{\theta}_m$) through the number of internal nodes $n$ and leaf nodes $n_\ell$. The priors on these continuous variables are defined to be,

$$\Delta_{m,j} \sim \text{Dir}(\boldsymbol{\alpha}), \qquad \tau_{m,j} \sim \mathcal{B}(1,1), \qquad \mu_{m,k} \sim \mathcal{N}(\alpha_\mu, \beta_\mu), \qquad \sigma_m \sim \Gamma^{-1}(\alpha_\sigma, \beta_\sigma),$$

for $\eta_j \in \mathcal{I}_m, \quad j = 1, \ldots, n$ and $\eta_{\ell_k} \in \mathcal{L}_m, \quad k = 1, \ldots, n_\ell$. Here, Dir denotes the Dirichlet distribution, $\mathcal{B}$ the beta distribution, $\mathcal{N}$ the normal distribution and $\Gamma^{-1}$ the inverse-gamma distribution. For regression problems, $\mu_{m,k}$ denotes the mean value of the assumed normal distribution within each leaf node $\eta_{\ell_k} \in \mathcal{L}_m$ and $\sigma_m$ is the assumed constant variance across all leaf nodes for tree subspace $m$.

### LIKELIHOOD DEFINITIONS

The likelihood for soft classification trees is taken to be the Dirichlet-Multinomial joint compound as defined as,

$$\ell(\mathbf{Y} \mid \mathbf{X}, \Theta_m, m) = \prod_{k=1}^{n_\ell} \left[ \frac{\Gamma(A)}{\Gamma(\Phi_k + A)} \prod_{c=1}^{C} \frac{\Gamma(\boldsymbol{\phi}_{c,k} + \alpha_m)}{\Gamma(\alpha_m)} \right] \tag{8}$$

where $C$ refers to the number of output classes, $n_\ell$ is the number of leaf nodes in the tree structure $m$ and $\Phi_k = \sum_c \boldsymbol{\phi}_{c,k}$ for each leaf $\eta_{\ell_k}$. The variable $\boldsymbol{\phi}_{c,k}$ represents the probability of each datapoint $(\mathbf{x}_i, y_i)$, for which the output class is $y_i = c$, being assigned to leaf node $\eta_{\ell_k}$ in tree $\mathbb{T}$ and is expressed as $\boldsymbol{\phi}_{c,k} = \sum_{i=1}^{N} \phi_{i,m,k}(\mathbf{x}_i \mid \mathbb{T}, \eta_{\ell_k}) \mathbb{I}(y_i = c)$ where $\phi_{i,m,k}$ is as previously defined in Equation 5.

Following Linero and Yang (2018), the likelihood for regression trees is defined to be,

$$\ell(\mathbf{Y} \mid \mathbf{X}, \Theta_m, m) = \prod_{i=1}^{N} \left(2\pi\sigma^2\right)^{-\frac{1}{2}} \times \exp\left[ -\frac{1}{2\sigma^2} \left(\sum_{k=1}^{n_\ell} \phi_{i,m,k}(\mathbf{x}_i) \cdot (\mu_{m,k} - y_i)\right)^2 \right] \tag{9}$$

where again $\phi_{i,m,k}$ is as defined in Equation 5.

## 3. DCC-Tree Sampling Algorithm

The DCC-Tree algorithm is based on the idea that the overall parameter space, which is defined by a joint discrete-continuous distribution, can be broken up into subspaces depending on the value of the discrete variable. The method progresses by considering a subset of all parameter values at each iteration. See Appendix A for details on the correctness of the overall strategy.

### 3.1. Overall Algorithm

The overall DCC-Tree sampling algorithm is shown in Algorithm 1. The method starts by generating an initial set of tree topologies $\mathcal{T}_m$ from the prior distribution, keeping track of those proposed and the corresponding number of times proposed $C_m$. Any tree topologies that have been selected more than a user-specified threshold $C_0$ are then stored in the set of currently active trees. The sampling method is run to initialise any new active tree for

each of the $N_c$ independent parallel chains using $N_{\text{INIT}}$ burn-in samples. This initialises the sampling method by adapting hyperparameters, such as the mass matrix and step size, and also provides a good starting point for subsequent local inference. The active tree with the highest utility value is then selected for local inference, in which an additional $N_s$ samples are generated for each of the $N_c$ parallel chains. Using the total number of local inference samples $N_T$ (including the new $N_s$ samples) the marginal likelihood of the current tree can be computed.

After local inference is complete, a global updating step is performed in which a new tree topology is proposed based on the current topology. If tree topology is already in the set of discovered trees, then the number of times the tree has been proposed $C_m$ is incremented, otherwise, it is added to the set. The algorithm continues to run for $T$ global iterations after which a set of samples that approximates the local posterior distribution and an estimate of the log marginal likelihood for each explored subspace is returned. The remainder of this section will provide specific details for the exploration of tree structures, marginal likelihood calculation and local density estimation components of the algorithm.

---

**Algorithm 1:** DCC-Tree Sampling Algorithm

---

**Input:** No. iterations $T$, no. initial trees $T_0$, no. parallel chains $N_c$, no. local samples
      per iteration $N_s$, times proposed threshold $C_0$, no. burn-in samples $N_{\text{INIT}}$.

**Init:** Set of discovered trees $\mathbb{D} = \emptyset$, set of currently active trees $\mathbb{A} = \emptyset$.

Generate $T_0$ trees from the prior and store $\mathbb{D} = \{\mathbb{T}_m\}$.

  **for** $t = 0$ **to** $T$ **do**

    **if** *any* $\mathbb{T}_m \in \mathbb{D}$ *selected more than* $C_0$ *times* **then**

      Add $\mathbb{T}_m$ to set of active trees $\mathbb{A}$.

    **if** *any new* $\mathbb{T}_m \in \mathbb{A}$ **then**

      Run $N_{\text{INIT}}$ burn-in samples for each of the $N_c$ parallel chains via standard
      adaption phase procedure (increasing $N_{init}$ as necessary).

    1. Calculate utility $U_m$ for each $\mathbb{T}_m \in \mathbb{A}$ via Equation 10 – select $\mathbb{T}_m$ with
    highest $U_m$ to perform local inference.

    2. Generate $N_s$ samples for each $N_c$ parallel chain for the selected tree $\mathbb{T}_m$.

    3. Calculate marginal likelihood estimate $\hat{Z}_m$ of tree $\mathbb{T}_m$ via Equation 11.

    4. Apply global update to selected $\mathbb{T}_m$ to propose $\mathbb{T}^*$.

    **if** $\mathbb{T}^* \in \mathbb{D}$ **then**

      Increase by one the number of times selected $C_{m^*}$ for $\mathbb{T}^*$.

    **else**

      Add $\mathbb{T}^*$ to $\mathbb{D}$.

    **end**

  **end**

  **for** $m = 0$ **to** $M = len(\mathbb{A})$ **do**

    Compute estimate of $\hat{p}_m(\Theta_m)$ via Equation 14.

  **end**

**Output:** Generated samples and marginal-likelihood estimates $\{\hat{p}_m(\Theta_m), \hat{Z}_m\}_{m=1}^M$.

---

### 3.2. Exploring Tree Structures

A challenging aspect of applying Bayesian inference to decision trees is adequately exploring the different tree structures. It is desirable to spend most of the computational effort considering trees with high posterior probability in order to better approximate the overall density. The standard grow/prune/stay random walk method is used to propose new tree topologies to be considered, such that there is a non-zero probability of exploring any topology. When a tree topology is first considered, the No-U-Turn-Sampler (NUTS) algorithm (Hoffman and Gelman, 2014) is run to initialise the sampling method for that topology. After the burn-in phase, a set of samples is collected, again via NUTS, and used to compute the marginal likelihood estimate $\hat{Z}_m$. This in turn is used in the calculation used to select the next tree for local inference.

The chosen function by which to select the next tree originates from Rainforth et al. (2018) and is defined for tree subspace $m$ as,

$$U_m := \frac{1}{S_m} \left( \frac{(1-\delta)\hat{\tau}_m}{\max_m\{\hat{\tau}_m\}} + \frac{\delta\hat{\rho}_m}{\max_m\{\hat{\rho}_m\}} + \frac{\beta \log \sum_m S_m}{\sqrt{S_m}} \right) \tag{10}$$

where $S_m$ is the number of times local inference has been performed on tree subspace $m$; $\hat{\tau}_m$ denotes the exploitation term; $\hat{\rho}_m$ is the exploration term; Calculations of these terms are described further in Appendix B.2. There are also two user-specified parameters: $0 \leq \delta \leq 1$ is a hyperparameter controlling the trade-off between exploration and exploitation; $\beta > 0$ is the standard optimism boost hyper-parameter (Zhou et al., 2020).

### 3.3. Marginal Likelihood Calculation

In the DCC-Tree algorithm, samples generated within each individual tree subspace provide an estimate of the local distribution. However, each subspace may not be equally important to the overall parameter space and must be assigned an associated weight. This weighting is related to the marginal likelihood of the local distribution. The method by which the marginal likelihood is estimated using MCMC-based samples is discussed here.

LAYERED ADAPTIVE IMPORTANCE SAMPLING
The marginal likelihood for each tree topology is estimated via the IS-after-MCMC method, layered adaptive importance sampling (see Section 5.4 of Llorente et al. (2023) for further details). Intuitively, this method attempts to create pseudo-samples to be used as importance samples from which the marginal likelihood can be approximated.

Consider a specific tree subspace $m$. Let the set of samples generated via HMC for this subspace be denoted $\nu_m^{(i,j)} = \{\boldsymbol{\Delta}_m^{(i,j)}, \boldsymbol{\tau}_m^{(i,j)}, \boldsymbol{\theta}_m^{(i,j)}\}$ for $i = 1, \ldots, N_T$ total number samples for each $j = 1, \ldots, N_c$ parallel chains. The estimate uses each sample $\nu_m^{(i,j)}$ to define a proposal distribution $q_{i,j,m}(\xi \mid \nu_m^{(i,j)}, \Sigma_j)$ from which $k = 1, \ldots, N_M$ pseudo-importance samples $\xi^{(i,j,k)}$ are drawn. Here, $\nu_m^{(i,j)}$ acts as the mean value and $\Sigma_j$ as a covariance matrix. These pseudo-importance samples are used to produce an estimate of the marginal likelihood via

$$\hat{Z}_m = \frac{1}{N_T N_c N_M} \sum_{i=1}^{N_T} \sum_{j=1}^{N_c} \sum_{k=1}^{N_M} \tilde{w}_m^{(i,j,k)}, \tag{11}$$

where $\tilde{w}_m^{(i,j,k)} = \frac{\tilde{p}_m(\xi^{(i,j,k)})}{\Phi_m(\xi^{(i,j,k)})}$. Here, $\tilde{p}_m(\xi^{(i,j,k)})$ is the unnormalised posterior distribution on parameters for tree subspace $m$, evaluated at each pseudo-importance sample $\xi^{(i,j,k)}$. The term $\Phi_m(\xi^{(i,j,k)})$ is also computed using these pseudo-importance samples (see Appendix B.1 for details), and is taken here to be the spatial definition,

$$\Phi_m(\xi^{(i,j,k)}) = \frac{1}{N_T} \sum_{n=1}^{N_T} q_{n,j,m}(\xi^{(i,j,k)} \mid \nu_m^{(n,j)}, \Sigma_j), \tag{12}$$

where the proposal distributions $q$ will be discussed next.

PROPOSAL DISTRIBUTIONS

The proposal density is defined jointly for all parameters within a given tree topology. Due to the possible multi-modality of the posterior distribution (more discussion on this in Section 3.4), the covariance of each proposal density is calculated only on the within-chain samples. The proposal distribution is defined in the unconstrained space to ensure valid proposals. The pseudo-importance samples for each parameter are drawn from a multivariate normal distribution centered around the original samples $\nu_m^{(i,j)}$ transformed to the unconstrained space,

$$\xi^{(i,j,k)} \sim \text{MVN}\left(f\left(\nu_m^{(i,j)}\right), \Sigma_j\right). \tag{13}$$

Here, MVN denotes the multivariate-normal distribution, $\Sigma_j$ is the parameter covariance of chain $j$ and $f$ is the transformation to the unconstrainted space.

## 3.4. Local Density Estimation

The NUTS algorithm uses the burn-in phase to adapt relevant hyperparameters to the sampling algorithm, while simultaneously encouraging movement to areas of high likelihood. However, it is possible that after this burn-in phase, samples corresponding to different chains within the same tree subspace may explore different modes. As a result, there is a potential multi-modality of each local parameter distribution $p_m(\Theta_m)$. Furthermore, these modes may have different posterior masses within the subspace, motivating the use of pseudo-importance samples with normalised weights. Equation 14 shows how the weighted importance samples can be used to approximate the posterior for tree subspace $m$,

$$\hat{p}_m(\Theta_m) = \sum_{i=1}^{N_T} \sum_{j=1}^{N_c} \sum_{k=1}^{N_M} w_m^{(i,j,k)} \delta_{\xi^{(i,j,k)}}(\cdot), \tag{14}$$

where $w_m^{(i,j,k)} = \frac{\tilde{w}_m^{(i,j,k)}}{\sum_{i=1}^{N_T} \sum_{j=1}^{N_c} \sum_{k=1}^{N_M} \tilde{w}_m^{(i,j,k)}}$, $\delta(\cdot)$ is the Dirac delta function and the unnormalised weights $\tilde{w}_m^{(i,j,k)}$ are as defined in Equation 11.

## 4. Experiments

The DCC-Tree algorithm was tested on a range of synthetic and real-world datasets commonly used in the Bayesian decision tree and machine learning literature. Each method was compared based on either the mean-square error or the accuracy across the testing and training datasets for regression and classification problems respectively.

### 4.1. Bayesian Tree Synthetic Datasets

The DCC-Tree algorithm was tested against two synthetic datasets common in the Bayesian decision tree literature, namely, the datasets from Chipman et al. (1998) and Wu et al. (2007), which will be referred to as CGM and WU respectively. Details of these datasets are presented in Appendix C. The DCC-Tree algorithm was run for $T = 500$ iterations, with $N_s = 100$ local inference samples per iteration. Each new tree subspace was initialised using $N_{\text{INIT}} = 2000$ burn-in samples for the WU dataset and $N_{\text{INIT}} = 5000$ for the CGM dataset. For each, $N_M = 10$ pseudo-importance samples were drawn to compute the marginal likelihood estimate. The DCC-Tree method was run 10 times for different initialisation values with metrics averaged across all runs. A relatively non-informative prior on the tree structure ($\alpha_{\text{SPLIT}} = 0.95$, $\beta_{\text{SPLIT}} = 1.0$), with the values of the split hyperparameters $h_{\text{INIT}}$ and $h_{\text{FINAL}}$ summarised in Table 5 in Appendix D. The results for the DCC-Tree algorithm and comparison to other methods for the WU and CGM datasets are shown in Table 1.

| | | CGM | WU | HMC-DF | HMC-DFI | DCC-Tree |
|---|---|---|---|---|---|---|
| CGM | Train MSE | 0.043(1.9e-4) | **0.042(5.8e-5)** | 0.043(2.0e-4) | 0.043(1.9e-4) | 0.043(5.2e-5) |
| | Test MSE | 0.064(0.014) | 0.062(0.001) | 0.041(4.6e-4) | 0.041(4.5e-4) | **0.040(1.4e-4)** |
| WU | Train MSE | 0.059(2.3e-4) | **0.054(1.4e-3)** | 0.060(7.4e-4) | 0.060(4.4e-4) | 0.058(3.3e-5) |
| | Test MSE | 0.112(0.034) | 0.073(0.038) | 0.059(2.1e-3) | 0.060(2.5e-3) | **0.059(5.3e-4)** |

Table 1: Testing and training MSE for various methods for the synthetic datasets of Chipman et al. (1998) and Wu et al. (2007).

The DCC-Tree algorithm exhibits the best testing performance across the different methods for both datasets, although only just better than the other HMC-based methods. Notably, the variance of the DCC-Tree method is lower than other methods, and in some cases, by nearly an order of 10. Both CGM and WU methods show signs of overfitting on the two datasets, clearly noted when comparing the difference in training and testing performance. In fact, the WU method performs best on the training data for both datasets but is much worse than all HMC-based methods with respect to the testing performance.

The marginal posterior distribution on the tree structure was also visualised for the DCC-Tree method, as defined by the marginal likelihood estimate $\hat{Z}_m$ for each tree subspace. Figure 1 shows this for both the WU and CGM datasets for a single run, with the label for the true tree structure highlighted in red. Using a relatively non-informative prior on the tree structure meant that the marginal likelihood estimate was affected mainly by the likelihood. However, Figure 1 does illustrate that this can be an issue when considering the marginal likelihood plot for the CGM dataset. The tree structure used to generate the data has a very low posterior probability. Instead, a tree structure with the same number of leaf nodes and which provides the same partition of the data (see Figure 4(b) in the appendix) has a posterior probability close to one. Further investigation into this issue showed that none of the parallel chains used for local inference discovered the correct mode for the true tree, resulting in a much lower marginal likelihood.

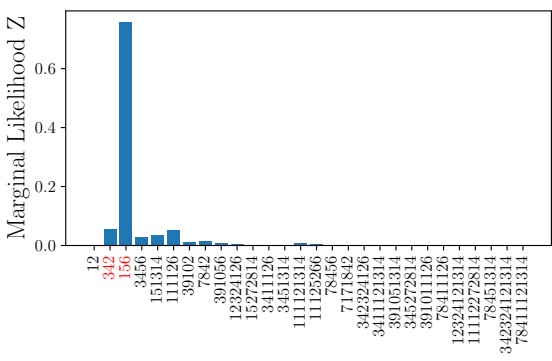 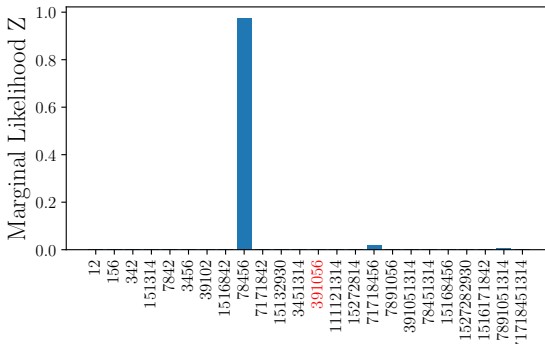

(*a*) Marginal likelihood for the WU dataset.  (*b*) Marginal likelihood for the CGM dataset.

Figure 1: Marginal likelihood of each tree structure for the DCC-Tree method. The tree structure used to generate the data is highlighted in red.

It is straightforward to plot the posterior predictive distribution for both the RJHMC-Tree and DCC-Tree methods as leaf parameters are not marginalised out and are therefore easily accessible for each method. For both the CGM and WU datasets, two testing data-points were randomly selected for which to compute the posterior predictive distributions. Figure 2 shows the posterior predictive distribution for the randomly selected testing data-points. The predictive distributions are compared to the predicted output from the standard CART model and the true output. It can be seen that in all cases, the distributions produced by the DCC-Tree method line up nearly exactly with the RJHMC-Tree methods. Furthermore, each strongly predicts the true value, in that the posterior distributions are highly peaked around the true output.

### 4.2. Real-world Datasets

The DCC-Tree method was also tested against a range of real-world datasets: Iris, Breast Cancer Wisconsin (Original), Wine, and Raisin Datasets (all available from Dua and Graff (2017)) and compared to other Bayesian decision tree methods. The DCC-Tree method was run for $T = 500$ iterations with $N_s = 100$ local inference samples generated per iteration. Split hyperparameters used for each dataset are summarised in Table 5 in Appendix D with a relatively uninformative prior on the tree structure again used. Training and testing metrics were averaged across 10 runs of the DCC-Tree method with different initialisation values. The results are shown in Table 2, along with the results for the other Bayesian tree methods. The standard deviation across the different runs is presented in parentheses. The best-performing method for each dataset is shown in bold.

The DCC-Tree method was also compared to other HMC-based samplers with respect to computational complexity. Table 6 in the appendix shows that the DCC-Tree method, which performs similarly to other HMC-based methods in terms of performance metrics, exhibits substantially improved computational efficiency across nearly all synthetic and real-world datasets.

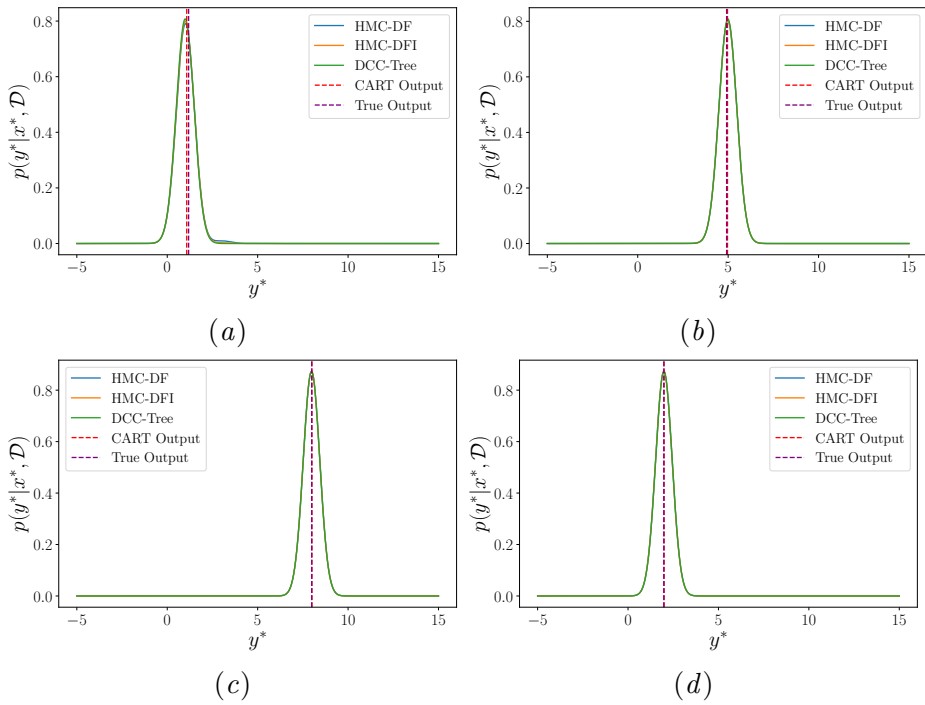

Figure 2: Posterior predictive distributions for randomly selected testing datapoints from the synthetic WU (a),(b) and CGM (c),(d) datasets.

|  |  | CGM | SMC | WU | HMC-DF | HMC-DFI | DCC-Tree |
|---|---|---|---|---|---|---|---|
| BCW | Train Acc. | 0.983(0.004) | **0.987(0.004)** | 0.978(0.005) | 0.973(0.005) | 0.981(0.003) | 0.982(0.001) |
|  | Test Acc. | 0.939(0.014) | 0.924(0.010) | 0.922(0.017) | 0.940(0.010) | 0.952(0.007) | **0.952(0.004)** |
| Iris | Train Acc. | **0.985(0.007)** | 0.981(0.004) | N/A[1] | 0.977(0.010) | 0.975(0.009) | 0.981(0.000)[2] |
|  | Test Acc. | 0.908(0.022) | 0.909(0.022) |  | 0.906(0.026) | **0.917(0.023)** | 0.911(1.2e-16)[2] |
| Wine | Train Acc. | 0.957(0.016) | **0.985(0.011)** | N/A[1] | 0.949(0.021) | 0.952(0.016) | 0.958(0.010) |
|  | Test Acc. | 0.916(0.046) | **0.978(0.022)** |  | 0.950(0.039) | 0.948(0.022) | 0.958(0.020) |
| Raisin | Train Acc. | 0.864(0.007) | 0.863(0.004) | 0.862(0.007) | 0.866(0.003) | 0.864(0.005) | **0.867(0.001)** |
|  | Test Acc. | 0.843(0.010) | 0.842(0.010) | 0.843(0.012) | **0.847(0.004)** | 0.838(0.007) | 0.844(0.002) |

[1] Uses Binomial likelihood (only two output classes allowed).

[2] The DCC-Tree algorithm predicted similar class probabilities across each chain when applied to the Iris dataset, resulting in the small variance in the metrics as shown here.

Table 2: Comparison of metrics for various methods for different real-world datasets.

## 5. Discussion and Future Work

We have proposed a novel sampling method for Bayesian decision trees that incorporates the efficiency of HMC into the DCC framework to provide efficient and fast sampling. Motivated

by the work of Zhou et al. (2020) in the context of probabilistic programming, the tree-sampling problem has been reduced into its constituents such that local inference can be applied, and then appropriately combined to give an overall estimate of the distribution. The efficacy of the sampling method developed in this paper has been demonstrated on a range of synthetic and real-world datasets and compared to existing methods.

By associating a unique parameter vector to each tree topology, the DCC-Tree method successfully implements a more natural way of sampling from the posterior distribution. The space is divided based on the different tree topologies and local inference is conducted within each subspace, combining the samples to recover the overall posterior. The method keeps track of each considered tree topology such that the burn-in period is only required once per unique structure, reducing the per-proposal complexity of other HMC-based approaches. This also enables the full benefits of the HMC sampling routine to be exploited.

Comparing the performance on the synthetic datasets shows that the DCC-Tree algorithm gives the best testing performance out of all Bayesian tree methods (see Table 1), however, only just better than other HMC-based algorithms. The performance on real-world data sets, Table 2, show that the DCC-Tree method performs similarly to the other HMC-based methods, outperforming the non-HMC methods on all but one dataset. On top of the performance, one major benefit of the DCC-Tree method is consistency. This is best shown by considering the standard deviation of the test performance in both Table 1 and 2. For the synthetic datasets, the standard deviation values are nearly an order of ten better than the next best method. For real-world datasets, in nearly all cases, the standard deviation values are close to half that of any other method.

One clear benefit of the DCC-Tree method is the improvement in computational efficiency via reducing overhead, which could account for the dramatic reduction observed in Table 6. However, this could be attributed to the difference in implementations; RJHMC-Tree methods run one chain at a time, whereas the DCC-Tree method runs multiple parallel chains within each run. Nevertheless, the possibility to run parallel chains within the inference algorithm itself should be interpreted as a benefit of the DCC-Tree implementation.

However, there are some shortcomings to the DCC-Tree algorithm. Similar to existing methods, new tree topologies are discovered in a random-walk-like fashion, although some improvement has been made using the function in Equation 10. In addition to this, the estimated marginal likelihood of the true tree is similar to those that contain it as a subtree. As a result, this function does allocate significant exploration effort to these larger trees. Nonetheless, these trees could be considered just as good as the true tree if only considering predictive ability. Lastly, once a tree structure has been initialised, further sampling remains around the discovered area of the local posterior distribution. Although somewhat mitigated through the use of multiple chains, this problem remains, and is particularly evident when considering the CGM marginal likelihood estimates in Figure 1. Possible extensions of this work could attempt to address this issue by 'restarting' to discover possible new modes.

Overall, the DCC-Tree algorithm appears to have made a significant improvement in exploring the posterior distribution of the decision tree. By altering the underlying ideology, the approach has moved from one where the proposal scheme couples the tree structure and decision parameters to one where each distinct topology is uniquely associated with a parameter vector. This paper demonstrated that this way of exploring the Bayesian decision tree posterior improved performance and consistency compared to other inference methods.

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

## Appendix A. Validity of Method

### A.1. Problem Statement and Assumptions

An important prerequisite for the proof of correctness is ensuring that the division and recombination of the parameter space results in the correct overall posterior distribution. These assumptions and definitions required for this discussion will be presented now, starting with the parameter vector in Definition 1.

**Definition 1 (DCC-Tree Parameter Vector)** *Let $\mathcal{M}$ be a discrete random variable whose value specifies a unique tree structure. Further, let the random variable $\Theta_m \in \boldsymbol{\Theta}_m \subset \mathbb{R}^{n_m}$ represent the associated parameters of each local tree subspace with appropriate dimension. Assume that there are $M$ topologies that span the entire parameter space such that $\mathcal{M} \in \{1, \ldots, M\}$. The DCC-Tree parameter vector is then defined as,*

$$\Theta = \begin{bmatrix} \Theta_1 & \Theta_2 & \cdots & \Theta_M \end{bmatrix} \in \boldsymbol{\Theta}, \tag{15}$$

*where*

$$\boldsymbol{\Theta} = \prod_{m=1}^{M} \boldsymbol{\Theta}_m \subset \mathbb{R}^{n_\Theta}, \quad n_\Theta = \sum_{m=1}^{M} n_m. \tag{16}$$

Careful consideration needs to be made as to the precise definition of the sample space. First, it is assumed that each subspace $\boldsymbol{\Theta}_i$ has compact support. Second, each parameter subspace is disjoint and covers the entire space, i.e. $\boldsymbol{\Theta}_i \cap \boldsymbol{\Theta}_j = \emptyset$. The overall parameter space includes both the parameter vector $\Theta$, which represents the continuous variables, and the discrete variable $\mathcal{M}$, as is shown in Definition 2.

**Definition 2 (DCC-Tree Parameter Space)** *The DCC-Tree joint parameter-index vector* $(\Theta, \mathcal{M})$ *is defined on the mixed discrete-continuous space,*

$$\mathbb{S} := \mathbb{R}^{n_\Theta} \times \mathbb{Z}. \tag{17}$$

Assumption 3 defines how the output distribution can be simplified when given the tree subspace $m$.

**Assumption 3** *It is assumed that given the subtree under consideration, the output can be modelled as,*

$$p(y \mid x, \Theta, \mathcal{M} = m) \equiv p(y \mid x, \Theta_m). \tag{18}$$

*where $\mathcal{M}$ denotes the discrete random variable representing the tree topology under consideration. That is to say, given that the value of the discrete variable is $m$, the output distribution depends only on the corresponding random variable $\Theta_m$.*

Note that the remainder of this section will use the simplification in notation whereby when the discrete variable takes on a specific value then the discrete random variable will be dropped, i.e. instead of $\mathcal{M} = m$ is simplified to just $m$.

The following assumption defines an important property of the sample space once a specific subspace has been selected. In particular, it assumes that the space corresponding to the remaining parameter vector is uniform over the compact space, such that it integrates to one. This is necessary for the last theorem of this section, which will show that under this assumption, only local samples are required for the approximation of the posterior predictive distribution.

**Assumption 4** *Let $\Theta_{\backslash m} \in \boldsymbol{\Theta}_{\backslash m}$ denote the component of the parameter vector defined by $\Theta \backslash \Theta_m$. When considering subspace $m$, the distribution of $\Theta_{\backslash m}$ is assumed to be uniform over the compact space $\boldsymbol{\Theta}_{\backslash m}$. That is,*

$$p(\Theta_{\backslash m} \mid m) = c_m \tag{19}$$

*where the value of $c_m$ is such that*

$$\int_{\boldsymbol{\Theta}_{\backslash m}} p(\Theta_{\backslash m} \mid m) d\Theta_{\backslash m} = 1. \tag{20}$$

Assumption 4 enables simplification of the joint prior distribution across the continuous random variables $\Theta$ and the discrete random variable $\mathcal{M}$, as is discussed in the following lemma.

**Lemma 5 (Prior Distribution)**  *If the discrete random variable takes on the value $m$, the joint prior distribution $p(\Theta, m)$ can be expressed as,*

$$p(\Theta, m) = c_m p(\Theta_m \mid m) p(m). \tag{21}$$

**Proof**  *Through the repeated application of conditional probability, the joint prior distribution can be written as,*

$$p(\Theta, m) = p(\Theta \mid m) p(m) \tag{22}$$
$$= p(\Theta_m \mid m) p(\Theta_{\backslash m} \mid m) p(m) \tag{23}$$
$$= c_m p(\Theta_m \mid m) p(m) \tag{24}$$

*where the second line is due to the independence of $\Theta_m$ and $\Theta_{\backslash m}$ and the third follows from Assumption 4.* ∎

The purpose of this method is to approximate the posterior distribution such that expectations with respect to this distribution can be evaluated. Given a dataset $\mathcal{D} = \{\mathbf{X}, \mathbf{Y}\}$, this corresponds to generating samples from the joint posterior distribution $p(\Theta, \mathcal{M} \mid \mathcal{D})$. The challenge with drawing samples from this joint posterior is that it is a mixed continuous-discrete distribution. However, once a particular subspace has been selected, inference on that subset of parameters is straightforward. The remainder of this section will discuss the connection between the posterior distribution $p(\Theta, m \mid \mathcal{D})$ and the distribution on the local parameters $p(\Theta_m \mid \mathcal{D}, m)$. This discussion begins by first introducing some notation that will simplify the later theorems.

**Definition 6**  *Let the discrete random variable take on the value $m$. Then the following notation is defined with respect to the distribution $p(\Theta_m \mid \mathcal{D}, m)$,*

$$p_m(\Theta_m) \triangleq p(\Theta_m \mid \mathcal{D}, m) \tag{25}$$
$$Z_m \triangleq p(\mathbf{Y} \mid m, \mathbf{X}). \tag{26}$$

*Note that this implies the following relationship,*

$$p_m(\Theta_m) Z_m = p(\mathbf{Y} \mid \Theta_m, \mathbf{X}) p(\Theta_m \mid m). \tag{27}$$

The following theorem provides the relationship between the posterior distribution $p(\Theta, m \mid \mathcal{D})$ and the distribution on the local parameters $p(\Theta_m \mid \mathcal{D}, m)$.

**Theorem 7 (Posterior Distribution)**  *Given a dataset $\mathcal{D}$, the posterior distribution can be expressed with respect to the local parameter distribution as,*

$$p(\Theta, m \mid \mathcal{D}) = \bar{w}_m \, c_m \, p_m(\Theta_m). \tag{28}$$

*where*

$$\bar{w}_m = \frac{Z_m p(m)}{\sum_{m=1}^{M} Z_m \, p(m)} \tag{29}$$

**Proof** *The application of Bayes Theorem to the posterior distribution gives the following,*

$$p(\Theta, m \mid \mathcal{D}) = \frac{p(\mathbf{Y} \mid \Theta, m, \mathbf{X}) \, p(\Theta, m \mid \mathbf{X})}{p(\mathbf{Y} \mid \mathbf{X})}, \tag{30}$$

$$= \frac{p(\mathbf{Y} \mid \Theta_m, \mathbf{X}) \, p(\Theta_m \mid m) \, p(m) \, c_m}{p(\mathbf{Y} \mid \mathbf{X})}, \tag{31}$$

$$= \frac{p_m(\Theta_m) \, Z_m \, p(m) \, c_m}{p(\mathbf{Y} \mid \mathbf{X})}, \tag{32}$$

*where the second line follows from Equation 21, and the third line from Equation 27. Using the law of total probability, the denominator can be expressed as,*

$$p(\mathbf{Y} \mid \mathbf{X}) = \sum_{m=1}^{M} p(\mathbf{Y}, m \mid \mathbf{X}) \tag{33}$$

$$= \sum_{m=1}^{M} p(\mathbf{Y} \mid m, \mathbf{X}) \, p(m) \tag{34}$$

$$= \sum_{m=1}^{M} Z_m \, p(m). \tag{35}$$

*Substitution into Equation 32 gives the required expression,*

$$p(\Theta, m \mid \mathcal{D}) = \frac{Z_m \, p(m)}{\sum_{m=1}^{M} Z_m \, p(m)} \, c_m \, p_m(\Theta_m) \tag{36}$$

$$= \bar{w}_m \, c_m \, p_m(\Theta_m), \tag{37}$$

*with the normalised weights defined as,*

$$\bar{w}_m = \frac{Z_m p(m)}{\sum_{m=1}^{M} Z_m \, p(m)}. \tag{38}$$

■

Theorem 7 shows that the parameter space for the DCC-Tree algorithm can be thought of as a distribution where, once a subspace has been selected, only the corresponding subset of the overall parameter vector is important. The only concern is the presence of the constant $c_m$ which is difficult to compute. Fortunately, as will be shown in the following theorem, this is not of concern when evaluating expectation integrals due to the cancellation of the term.

**Theorem 8 (Estimation of Posterior Predictive Distribution)** *Given a new data-point $\{\mathbf{x}^*, y^*\}$, the posterior predictive distribution $p(y^* \mid \mathbf{x}^*, \mathcal{D})$ can be approximated using samples from the joint posterior distribution $p(\Theta, m \mid \mathcal{D})$ as follows,*

$$p(y^* \mid \mathbf{x}^*, \mathcal{D}) \approx \frac{1}{L} \sum_{i=1}^{L} p(y^* \mid \mathbf{x}^*, \Theta_{m^i}^i), \qquad (m^i, \Theta_{m^i}^i) \overset{i.i.d.}{\sim} \bar{w}_m \, p_m(\Theta_m). \tag{39}$$

**Proof** *Recall that the posterior predictive distribution is an expectation with respect to some distribution. In this case, this is the joint discrete-continuous distribution, with the expectation defined to be,*

$$p(y^* \mid \mathbf{x}^*, \mathcal{D}) = \sum_{m=1}^{M} \int_{\Theta} p(y^* \mid \mathbf{x}^*, \Theta, m, \mathcal{D}) \, p(\Theta, m \mid \mathcal{D}) \, d\Theta \tag{40}$$

*Using Assumption 3, Assumption 4 and the result from Theorem 7, this expression can be simplified in the following manner,*

$$p(y^* \mid \mathbf{x}^*, \mathcal{D}) = \sum_{m=1}^{M} \int_{\Theta_m} \int_{\Theta_{\backslash m}} p(y^* \mid \mathbf{x}^*, \Theta, m, \mathcal{D}) \, p(\Theta, m \mid \mathcal{D}) \, d\Theta_{\backslash m} \, d\Theta_m \tag{41}$$

$$= \sum_{m=1}^{M} \int_{\Theta_m} \int_{\Theta_{\backslash m}} p(y^* \mid \mathbf{x}^*, \Theta_m) \, \bar{w}_m \, c_m \, p_m(\Theta_m) \, d\Theta_{\backslash m} \, d\Theta_m \tag{42}$$

$$= \sum_{m=1}^{M} \int_{\Theta_m} p(y^* \mid \mathbf{x}^*, \Theta_m) \, \bar{w}_m \, p_m(\Theta_m) \, d\Theta_m \tag{43}$$

*where it has been used that $y^*$ is independent of $\mathcal{D}$ given the parameter vector $\Theta$. The final expression shows that the posterior predictive distribution can be evaluated by considering only the weighted local distribution for each subspace $m$. This means that the integral can be approximated using Monte Carlo integration via*

$$p(y^* \mid \mathbf{x}^*, \mathcal{D}) \approx \frac{1}{L} \sum_{i=1}^{L} \frac{p(y^* \mid \mathbf{x}^*, \Theta_{m^i}^i) \, \bar{w}_{m^i} \, p_{m^i}(\Theta_{m^i}^i)}{q(m^i, \Theta_{m^i}^i)}, \qquad (m^i, \Theta_{m^i}^i) \overset{i.i.d.}{\sim} q(m, \Theta_m) \tag{44}$$

*where $q(\cdot)$ is a user-specified proposal distribution. In particular, this proposal can be taken as*

$$q(m, \Theta_m) \triangleq \bar{w}_m \, p_m(\Theta_m) \tag{45}$$

*which results in the following simplification*

$$p(y^* \mid \mathbf{x}^*, \mathcal{D}) \approx \frac{1}{L} \sum_{i=1}^{L} p(y^* \mid \mathbf{x}^*, \Theta_{m^i}^i) \qquad (m^i, \Theta_{m^i}^i) \overset{i.i.d.}{\sim} \bar{w}_m \, p_m(\Theta_m), \tag{46}$$

*giving the required expression.* ∎

It is clear from Theorem 7 that, when a particular subspace $\Theta_m$ is under consideration, the joint posterior simplifies down to the weighted local distribution. Further, Theorem 8 shows that only samples from each parameter subspace $\Theta_m$ are required to generate the estimate of $p_m(\Theta_m)$ and therefore approximate integrals regarding the overall distribution. It was important to establish these concepts before continuing to the next section, which details the correctness of the method by considering the sampling method to be broken up into local and global components.

### A.2. Correctness of Method

The correctness of the overall DCC-Tree sampling method relies on two parts: the validity of the local inference method to converge to the local target posterior distribution and the consistency of the local density approximations to recover the overall distribution. The method is based on the idea that the sample space can be split up and approximated as follows,

$$p(\Theta \mid \mathcal{D}) = \sum_{m=1}^{M} p(\Theta, m \mid \mathcal{D}) = \sum_{m=1}^{M} \frac{c_m \, Z_m \, p(m)}{\sum_{m=1}^{M} Z_m \, p(m)} \, p(\Theta_m \mid m, \mathcal{D}) \tag{47}$$

$$\approx \sum_{m=1}^{M} \frac{c_m \, \hat{Z}_m \, p(m)}{\sum_{m=1}^{M} \hat{Z}_m \, p(m)} \, \hat{p}(\Theta_m \mid m, \mathcal{D}) \triangleq \hat{p}(\Theta \mid \mathcal{D}) \tag{48}$$

where $\hat{Z}_m$ and $\hat{p}(\Theta \mid m, \mathcal{D})$ are the approximations for $Z_m$ and $p(\Theta \mid m, \mathcal{D})$ respectively. Note that this shows that the distribution $p(\Theta \mid \mathcal{D})$ can not be expressed directly as it contains the term $c_m$, but as shown in Theorem 8 for the posterior predictive distribution, the samples generated from the local distribution $p(\Theta_m \mid m, \mathcal{D})$ can be used to evaluate expectation integrals with respect to this distribution. For the DCC-Tree algorithm, HMC is used to generate these local samples that can be used to provide these estimates.

HMC attempts to sample from a target distribution by using well-informed proposals that incorporate information about the likelihood distribution into the proposal scheme. For the DCC-Tree method, the target distribution is the distribution on the local parameters $p(\Theta_m \mid m, \mathcal{D})$. HMC draws samples from this distribution by using the relationship via Bayes' theorem as follows (note that the dependence on data on data $\mathcal{D} = \{\mathbf{X}, \mathbf{Y}\}$ is explicitly included here):

$$p(\Theta_m \mid m, \mathcal{D}) = \frac{p(\mathbf{Y} \mid \Theta_m, m, \mathbf{X}) p(\Theta_m \mid m)}{p(\mathbf{Y} \mid m, \mathbf{X})}. \tag{49}$$

Samples from the target distribution $p(\Theta_m \mid m, \mathcal{D})$ are generated in a way where only the unnormalised density (the numerator) is required due to the cancellation of terms in the accept/reject step of the algorithm Metropolis et al. (1953); Hastings (1970). Note that as a result of this, the denominator – which is referred to as the marginal likelihood – is also not required by the algorithm to produce the desired samples.

The correctness of using HMC as the local inference scheme results from the validity of the sample algorithm itself, as is described in Lemma 9.

**Lemma 9 (Validity of Local Inference (HMC))** *Given the discrete random variable takes on the value $m$, the HMC algorithm generates samples from the local distribution $p(\Theta_m \mid m, \mathcal{D})$ as defined by Equation 49.*

**Proof** *As is standard with HMC sampling methods, the potential function is aligned with the negative-log likelihood of the unnormalised target distribution. For the DCC-Tree algorithm, this is defined to be,*

$$U \triangleq -\log\left(p(\mathbf{Y} \mid \Theta_m, m, \mathbf{X}) \, p(\Theta_m \mid m)\right) \tag{50}$$

*where $p(\mathbf{Y} \mid \Theta_m, m, \mathbf{X})$ is defined by either Equation 8 for classification or Equation 9 for regression and $p(\Theta_m \mid m)$ by Equation 7. The standard application of HMC under this*

*potential energy definition then ensures that samples are generated from the normalised local distribution $p(\Theta_m \mid m, \mathcal{D})$.* ∎

The rest of this section focuses on the idea that, given the approximation techniques satisfy some assumptions, the estimated distribution can be recombined to give an estimate of the overall posterior distribution. Note that the rest of the section, including the next set of assumptions, follows the original proof provided in Appendix C of Zhou et al. (2020) but where the notation has been changed to reflect that used in this paper.

**Assumption 10** *It is assumed that the total number $M$ of tree subspaces $\Theta_m$ is finite.*

**Assumption 11** *It is assumed that the number of iterations $T$ required to find all tree subspaces is almost surely finite.*

**Assumption 12** *Every tree subspace $m \in \{1, \ldots, M\}$ has an associated local density estimate $\hat{p}(\Theta \mid m, \mathcal{D})$ that converges weakly in the limit of large numbers to the true distribution $p(\Theta \mid m, \mathcal{D})$ corresponding to that subspace. Further, each tree subspace $m$ has a local marginal likelihood estimate $\hat{Z}_m$ that converges in probability to the true marginal likelihood of that subspace $Z_m$.*

With the above assumptions, Theorem 13 proves the consistency of the DCC-Tree algorithm.

**Theorem 13 (Correctness of DCC-Tree Algorithm Zhou et al. (2020))** *If Assumptions 10 to 12 hold, then the estimate of the posterior density defined by Equation 48 generated by the DCC-Tree method converges weakly to the true distribution in the limit, that is,*

$$\hat{p}(\Theta \mid \mathcal{D}) \to p(\Theta \mid \mathcal{D}) \qquad as \qquad T \to \infty. \tag{51}$$

**Proof** *The proof follows that presented in Theorem 1 in Appendix C of the supplementary material for Zhou et al. (2020), but is presented here with the notation related to this paper.*

*For an arbitrary function $f$, the following result holds,*

$$\mathbb{E}_{\hat{p}(\Theta|\mathcal{D})}\left[f(\Theta)\right] = \int_{\Theta} f(\Theta)\,\hat{p}(\Theta \mid \mathcal{D})\,d\Theta \tag{52}$$

$$= \int_{\Theta} f(\Theta) \sum_{m=1}^{M} \frac{c_m\,\hat{Z}_m\,p(m)}{\sum_{m=1}^{M}\hat{Z}_m\,p(m)}\,\hat{p}(\Theta \mid m, \mathcal{D})\,d\Theta \tag{53}$$

$$= \sum_{m=1}^{M} \int_{\Theta} f(\Theta)\,\frac{c_m\,\hat{Z}_m\,p(m)}{\sum_{m=1}^{M}\hat{Z}_m\,p(m)}\,\hat{p}(\Theta \mid m, \mathcal{D})\,d\Theta \tag{54}$$

$$= \sum_{m=1}^{M} \int_{\Theta} f(\Theta)\,\frac{c_m\,Z_m\,p(m)}{\sum_{m=1}^{M}Z_m\,p(m)}\,p(\Theta \mid m, \mathcal{D})\,d\Theta \tag{55}$$

$$= \int_{\Theta} f(\Theta) \sum_{m=1}^{M} \frac{c_m\,Z_m\,p(m)}{\sum_{m=1}^{M}Z_m\,p(m)}\,p(\Theta \mid m, \mathcal{D})\,d\Theta \tag{56}$$

$$= \int_{\Theta} f(\Theta)\,p(\Theta \mid \mathcal{D})\,d\Theta \tag{57}$$

$$= \mathbb{E}_{p(\Theta|\mathcal{D})}\left[f(\Theta)\right] \tag{58}$$

*where Equation 55 is due to Slutsky's theorem. Therefore, the expectation of the function $f$ with respect to the approximated posterior distribution $\hat{p}(\Theta \mid \mathcal{D})$ converges to the expectation under the true distribution $p(\Theta \mid \mathcal{D})$. As $f$ was defined to be an arbitrary function then this result holds true in general, as required.* ∎

The next section will discuss the practical implementation of the DCC-Tree algorithm, and in particular, the methods for estimating the local distribution $\hat{p}(\Theta \mid m, \mathcal{D})$ and the marginal likelihood $\hat{Z}_m$ for each subspace $m$.

## Appendix B. DCC-Tree Implementation Details

Exact details on how certain calculations have been implemented are presented in this section.

### B.1. Log-Marginal Likelihood Calculation

From Section 3.3, the marginal likelihood estimate for tree $\mathbb{T}_m$ is calculated via

$$\hat{Z}_m = \frac{1}{N_T N_c N_M} \sum_{i=1}^{N_T} \sum_{j=1}^{N_c} \sum_{k=1}^{N_M} \tilde{w}_{i,j,k}^{(m)}, \quad \tilde{w}_{i,j,k}^{(m)} = \frac{\tilde{\pi}_m(\xi^{(i,j,k)})}{\Phi_m(\xi^{(i,j,k)})} \tag{59}$$

where $\pi_m(\xi^{(i,j,k)})$ is the posterior distribution on parameters and $\xi^{(i,j,k)}$ are the $k = 1, \ldots, N_M$ new parameter samples at iteration $i$ of chain $j$. The term $\Phi_m(\xi^{(i,j,k)})$ is computed using either the basic or spatial formulation,

$$\Phi_m(\xi^{(i,j,k)}) = q_{i,j,m}(\xi^{(i,j,k)} \mid \nu_m^{(i,j)}, \Sigma), \text{ or, } \Phi_m(\xi^{(i,j,k)}) = \frac{1}{N_T} \sum_{n=1}^{N_T} q_{n,j,m}(\xi^{(i,j,k)} \mid \nu_m^{(n,j)}, \Sigma) \tag{60}$$

These calculations are converted to log space as follows. The estimated log marginal likelihood is computed via

$$\log \hat{Z}_m = \text{logsumexp}\left(\log \tilde{w}_{i,j,k}^{(m)}\right) - \log(N_T N_c N_M) \tag{61}$$

where

$$\log \tilde{w}_{i,j,k}^{(m)} = \log \tilde{\pi}_m\left(\xi^{(i,j,k)}\right) - \log \Phi_m\left(\xi^{(i,j,k)}\right). \tag{62}$$

Access to $\log \pi_m\left(\xi^{(i,j,k)}\right)$ is available directly throughout the simulation. The term $\log \Phi_m\left(\xi^{(i,j,k)}\right)$ is calculated either as

$$\log \Phi_m\left(\xi^{(i,j,k)}\right) = \text{logsumexp}\left(\log q_{i,j,m}(\xi^{(i,j,k)})\right), \tag{63}$$

for the basic expression, or as the spatial version,

$$\log \Phi_m\left(\xi^{(i,j,k)}\right) = \text{logsumexp}\left(\log q_{n,j,m}(\xi^{(i,j,k)})\right) - \log(N_T), \tag{64}$$

where the proposal distribution $q$ are initialised in log-form as required (see Section 3.3 for details).

Lastly, the log-marginal likelihood does not need to be recalculated at each additional visit to the tree. Instead, it can be updated via,

$$\log \hat{Z}_m^{(t)} = \text{logsumexp}\left(\left[\log \hat{Z}_m^{(t-1)} + \log((t-1)N_s N_c N_M), \log w_{i,j,k}^{(t-1)}\right]\right) - \log(t N_s N_c N_M) \tag{65}$$

where $t \in \mathbb{N}$ denotes the visit number to that tree.

### B.2. Calculation of Exploitation Term

Through the simulation, the terms $\log(\hat{Z}_m)$ and $\log(\sigma_m^2)$ for each tree $\mathbb{T}_m$ are recorded. However, the value of these can be large, resulting in either 0 or an infinite value when taking the exponent and the loss of any relative information. Therefore, a numerically stable way is required to approximate the calculation of the exploitation term, which is computed as follows,

$$\frac{\hat{\tau}_m}{\max_m\{\hat{\tau}_m\}}, \quad \hat{\tau}_m = \sqrt{\hat{Z}_m^2 + (1+\kappa)\sigma_m^2}. \tag{66}$$

Only access to $\log(\hat{Z}_m)$ and $\log(\sigma_m^2)$ is provided as the method progresses. As such, the above expression for $\hat{\tau}_m$ is rewritten as

$$\sqrt{\hat{Z}_m^2 + (1+\kappa)\sigma_m^2} = \sqrt{\exp\left(2\log \hat{Z}_m\right) + (1+\kappa)\exp\left(\log \sigma_m^2\right)}. \tag{67}$$

Taking $A_m = \max\left\{2\log(\hat{Z}_m), \log(\sigma_m^2)\right\}$, the following transforms can be used to alter the above expression,

$$2\widetilde{\log(\hat{Z}_m)} = 2\log(\hat{Z}_m) - A_m \qquad\qquad \widetilde{\log(\sigma_m^2)} = \log(\sigma_m^2) - A_m$$

where $A_m$ is now large. This changes the expression in Equation 67 to

$$\sqrt{\hat{Z}_m^2 + (1+\kappa)\sigma_m^2} = \sqrt{\exp\left(A_m + 2\widetilde{\log \hat{Z}_m}\right) + (1+\kappa)\exp\left(A_m + \widetilde{\log \sigma_m^2}\right)} \tag{68}$$

$$= \underbrace{\sqrt{\exp(A_m)}}_{\text{unstable}} \cdot \underbrace{\sqrt{\exp\left(2\widetilde{\log \hat{Z}_m}\right) + (1+\kappa)\exp\left(\widetilde{\log \sigma_m^2}\right)}}_{\text{can be calculated}} \tag{69}$$

Since both the square root and exponential functions are monotonically increasing, the ratio in Equation 66 can be computed by evaluating the relative expression between tree topologies. To compute this ratio, assume that there are two tree topologies with subscripts $p$ and $q$ and assume without loss of generality that $q$ corresponds to the value for which $\hat{\tau}_m$ is maximum. The ratio for topology $p$ is then given by,

$$\frac{\hat{\tau}_p}{\max_m\{\hat{\tau}_m\}} = \sqrt{\exp(A_p - A_q)} \frac{\sqrt{\exp\left(2\widetilde{\log \hat{Z}_p}\right) + (1+\kappa)\exp\left(\widetilde{\log \sigma_p^2}\right)}}{\sqrt{\exp\left(2\widetilde{\log \hat{Z}_q}\right) + (1+\kappa)\exp\left(\widetilde{\log \sigma_q^2}\right)}} \tag{70}$$

where all terms are now numerically stable to calculate. This method also handles cases when either the log-marginal likelihood or log-variance is significantly larger than the other.

### B.3. Calculation of Exploration Term

The exploration term $\hat{\rho}_m$ as first described in Rainforth et al. (2018) is adopted here. This term is used to help define the possible improvement in the marginal likelihood estimation with additional inference samples. The expression relates to the log-weights of the current set of samples as follows,

$$\hat{\rho}_m \triangleq P\left(\hat{w}_m(T_a) > w_{th}\right) \approx 1 - \Psi_m(\log w_{th})^{T_a}, \tag{71}$$

and defines the probability that at least one new sample in a "look-ahead" horizon would have a log-weight greater than some threshold weight $w_{th}$. The function $\Psi_m$ is taken to be the cumulative distribution function for the normal distribution with mean and variance based on the current samples for tree $\mathbb{T}_m$. The value of the threshold weight $w_{th}$ is taken to be the maximum weight of the samples discovered so far across all active trees. The variable $T_a$ is a hyperparameter that represents the number of "look-ahead" samples in the horizon. As per Rainforth et al. (2018), a default value of $T_a = 1000$ is used for the DCC-Tree algorithm.

## Appendix C. True Tree Definitions for Synthetic Datasets

The synthetic dataset proposed by Wu et al. (2007) is defined such that the input space is given by,

$$\boldsymbol{x}_i = \begin{cases} (x_1, x_2, x_3) & \text{where } x_1 \sim \mathcal{U}_{[0.1,0.4]}, x_2 \sim \mathcal{U}_{[0.1,0.4]}, x_3 \sim \mathcal{U}_{[0.6,0.9]}, \text{ for } i = 1, \dots, 100, \\ (x_1, x_2, x_3) & \text{where } x_1 \sim \mathcal{U}_{[0.1,0.4]}, x_2 \sim \mathcal{U}_{[0.6,0.9]}, x_3 \sim \mathcal{U}_{[0.6,0.9]}, \text{ for } i = 101, \dots, 200, \\ (x_1, x_2, x_3) & \text{where } x_1 \sim \mathcal{U}_{[0.6,0.9]}, x_2 \sim \mathcal{U}_{[0.1,0.9]}, x_3 \sim \mathcal{U}_{[0.1,0.4]}, \text{ for } i = 201, \dots, 300, \end{cases}$$

where $\mathcal{U}_{[a,b]}$ represents the uniform distribution over the interval $[a, b]$. The corresponding outputs are then defined to be,

$$y = \begin{cases} 1 + \mathcal{N}(0, 0.25) & \text{if } x_1 \le 0.5 \text{ and } x_2 \le 0.5, \\ 3 + \mathcal{N}(0, 0.25) & \text{if } x_1 \le 0.5 \text{ and } x_2 > 0.5, \\ 5 + \mathcal{N}(0, 0.25) & \text{if } x_1 > 0.5, \end{cases}$$

with the two trees that are consistent with the data (equally as likely) shown in Figure 3.

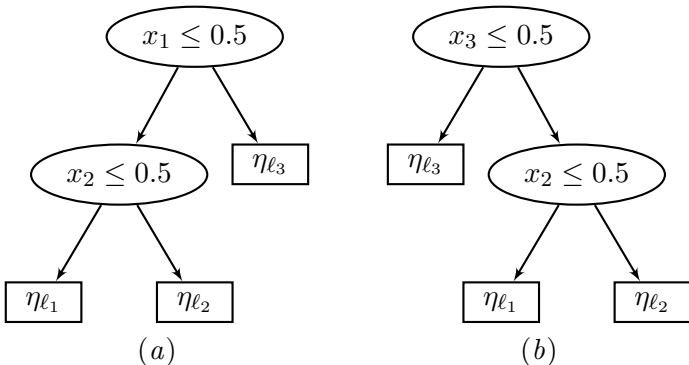

Figure 3: The two trees consistent with the synthetic dataset defined in Wu et al. (2007).

The synthetic dataset of Chipman et al. (1998) was adapted in the same manner as Cochrane et al. (2023). This differs from the original version by making the first split defined by a continuous input not categorical. All inputs are numerical with $n_x = 2$ and simulated via the tree structure shown in Figure 4($a$) to give $N_{\text{TRAIN}} = 800$, $N_{\text{TEST}} = 800$ datapoints. The variance is assumed constant across nodes with value $\sigma^2 = 0.2^2$. Note that Figure 4($b$) shows a tree with the same partition of the data as the true tree definition.

## Appendix D. Hyperparameters Optimisation

Hyperparameters used for the simulation of other Bayesian tree methods were taken from Cochrane et al. (2023). We provide analysis of an additional dataset in this paper - the synthetic dataset from Wu et al. (2007). Table 3 provides the values used for the grid search, with Table 4 detailing the final hyperparameters used for each of the other Bayesian decision tree methods. Note that the SMC method is not included as its implementation is

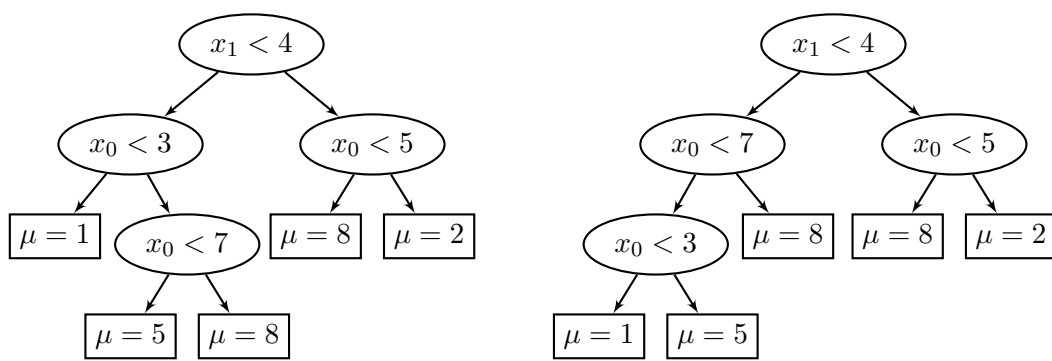

($a$) True tree definition used to generate dataset.

($b$) Tree with same partition of the data.

Figure 4: Tree representation of the synthetic dataset adapted from Chipman et al. (1998).

not applicable to regression datasets. The final set of hyperparameters used for the DCC-Tree method simulations is provided in Table 5. Note that hyperparameters not listed are default values.

| Method | Hyperparameters |
|---|---|
| WU | $\alpha = [0.5 : 0.5 : 4.0]$, $\beta = [0.5 : 0.5 : 4.0]$, $\mu_0 = [0 : 0.5 : 2]$, $n = [0.5 : 0.5 : 1.5]$, $\lambda = [8 : 1 : 10]$, $p = [0.3 : 0.2 : 0.7]$ |
| CGM | $\alpha = [0.5 : 0.5 : 4.0]$, $\beta = [0.5 : 0.5 : 2.5]$, $\mu_0 = [0 : 0.5 : 2]$, $n = [0.3 : 0.1 : 0.6]$, $\alpha_{\text{SPLIT}} = [0.45 : 0.25 : 0.95]$, $\beta_{\text{SPLIT}} = [1.0 : 0.5 : 2.5]$ |

Table 3: Grid search values considered in hyperparameter optimisation via 5-fold cross-validation for the synthetic dataset of Wu et al. (2007).

| Method | Hyperparameters |
|---|---|
| RJHMC | $h_{\text{INIT}} = 0.025$, $h_{\text{FINAL}} = 0.025$, $\alpha_{\text{SPLIT}} = 0.45$, $\beta_{\text{SPLIT}} = 2.5$ |
| WU | $\alpha = 4.0$, $\beta = 4.0$, $\mu_0 = 1.0$, $n = 1.0$, $\lambda = 10.0$, $p = 0.7$ |
| CGM | $\alpha = 3.5$, $\beta = 0.5$, $\mu_0 = 1.0$, $n = 0.5$, $\alpha_{\text{SPLIT}} = 0.95$, $\beta_{\text{SPLIT}} = 1.0$ |

Table 4: Final hyperparameters for the other Bayesian decision tree methods for the synthetic dataset of Wu et al. (2007).

| Dataset | Hyperparameters |
|---------|-----------------|
| BCW | $h_{\text{INIT}} = 0.1$, $h_{\text{FINAL}} = 0.025$ |
| CGM | $h_{\text{INIT}} = 0.01$, $h_{\text{FINAL}} = 0.001$ |
| Iris | $h_{\text{INIT}} = 0.01$, $h_{\text{FINAL}} = 0.01$ |
| Raisin | $h_{\text{INIT}} = 0.05$, $h_{\text{FINAL}} = 0.001$ |
| Wine | $h_{\text{INIT}} = 0.025$, $h_{\text{FINAL}} = 0.025$ |
| WU | $h_{\text{INIT}} = 0.5$, $h_{\text{FINAL}} = 0.025$ |

Table 5: Hyperparameters used for the DCC-Tree method for different datasets.

## Appendix E. Computational Comparison

Improvements in computational efficiency compared to other HMC-based methods was investigated with the results shown in Table 6. Note that both the time per proposal and the time per proposal with non-zero weight (and therefore contribution to the final posterior) are provided for the DCC-Tree method.

| | HMC-DF | HMC-DFI | DCC-Tree | DCC-Tree (non-zero weight) |
|---|--------|---------|----------|----------------------------|
| WU | 35.83 | 26.37 | 0.008576 | 0.0150 |
| CGM | 64.99 | 68.23 | 0.5738 | 47.61 |
| BCW | 32.61 | 18.28 | 0.00626 | 0.00981 |
| Iris | 24.03 | 15.26 | 0.00606 | 0.006369 |
| Raisin | 24.67 | 28.27 | 0.01313 | 0.0254 |
| Wine | 36.01 | 22.39 | 0.01381 | 0.0392 |

Table 6: Time (seconds) per proposal for different HMC-based methods, averaged across the 10 runs. Regression datasets are displayed first. Time includes any burn-in phase for the sampling method.

