# OpenReview forum: "Divide, Conquer, Combine Bayesian Decision Tree Sampling"
_approximateinference.org/AABI/2025/Proceedings_Track — AABI 2025 Proceedings Track_

### Official Review · Reviewer_PWeJ · 2025-02-25
**Additional details & experiments would strengthen the paper**

**Rating:** 4
**Confidence:** 5

**Review:**

The paper introduces a new sampling algorithm for a soft Bayesian treed regression. The method works by partitioning the space of regression trees into disjoint classes (of varying dimension) indexed by the tree topology. Within each class, NUTS is used to sample the vector of decision-rule and leaf output parameters. These local samples are then used to weight the individual subspaces and then a new tree structure is proposed.

On a few synthetic and real world datasets, the proposed DCC-Tree model yields better out-of-sample predictions than the hard decision tree models of Chipman et al. and Wu et al and HMC-based soft decision tree methods. While I found the proposal very interesting, I believe that there are some aspects that merit further comment/clarification and that the paper would be strengthened by more extensive experiments.

Non-identification: Just before Figure 1, you write that "the tree structure used to generate the data has a very low posterior probability. Instead, a tree structure with the same number of leaf nodes and which provides the same partition of the data has posterior probability close to one." I am confused by this phenomenon. If the trees induce the same partition of the data, shouldn't they yield the same likelihood value? If so, then the only way to see such a big discrepancy in posterior probability would be through the prior. But I'm a bit skeptical that this the only source of the discrepancy, as the two trees have the same number of nodes. Do the two trees have very different depths? It would be helpful to investigate the discrepancy further (e.g., by including a visualization of the two trees and decomposing the prior and likelihood values).

I suspect what's actually happened is that the Markov chain navigated to the vicinity of the non-data generating tree and was unable to navigate to the true data generating tree using the grow/change/move/swap proposals in the global update steps. This behavior is essentially similar what Chipman et al. (1998) report. On that view, it would seem like the DCC-Tree algorithm, while yielding more efficient exploration of decision-node and leaf-node parameters conditional on the tree structure, inherits exactly the same limitations of exploring across tree topologies as existing work.

Relatedly, on the view that the trees are non-identified, what is the utility of exploring the tree space well? Do you observe better mixing in the prediction space?

Comparison to a hybrid algorithm: you claim that DCC-Tree ``appears to have made a significant improvement in exploring the posterior distribution of the decision tree.'' How much of the improvement is due to the use of soft decision trees and how much is attributable to the algorithmic difference (i.e., using DCC instead of the standard grow/prune/change/swap). To probe this, I believe you need to compare to a hybrid algorithm that uses soft decision rules but uses the grow/prune/change/swap proposals. For such a hybrid, in grow moves, you would propose new decision-node parameters from the prior (which leads to considerable cancellation in the MH ratio). See, for instance, [Nguyen et al. (2024)](https://arxiv.org/abs/2411.08849), who do something similar in their implementation of BART with oblique (i.e., hyperplane) decision rules. A comparison to this hybrid would help readers understand why the new method works better than existing ones.

Comparison to ensemble methods: How does the runtime of DCC-Tree compare to the runtime of the CGM and WU algorithms? And how do all of the methods compare, in terms of predictions, to ensemble methods like BART and [Soft BART](https://arxiv.org/abs/2210.16375)? I think it would be helpful to include comparisons to (Soft) BART to set a benchmark: if the proposed method yields similar (or better) predictions to one of the ensemble methods in comparable (or faster) runtime, then that is a pretty impressive finding (since, in my experience, tree ensemble methods work better than single tree methods)!


Initialization: If the true regression function can only be approximated with a very deep tree, you will need to run your algorithm for a long time in order to grow the trees in $\mathbb{D},$ which were initialized from prior draws. Instead, have you considered initializing the tree topologies using a CART tree fit to a bootstrap re-sample (or otherwise randomly perturbed) dataset? This might allow you to start closer to higher posterior probability tree topologies.

---

### Official Review · Reviewer_MWNq · 2025-02-28

**Rating:** 7
**Confidence:** 4

**Review:**

This paper presents tree structures as disjoint subspaces (DCC-Tree) models, where each tree structure is represented separately, assigning unique parameters. It uses soft decision functions to replace complex splits with probabilistic paths for smoother sampling. Importance sampling and weighted subspace estimates are used to enhance the posterior reconstruction of the marginal likelihood estimation. An exploration-exploitation strategy prioritizes the most promising tree structures for efficient inference.

The experiments show that DCC-Tree outperforms HMC-based methods' accuracy and stability on synthetic datasets. It demonstrates competitive accuracy with lower variance on real-world datasets like Iris and Breast Cancer Wisconsin. Additionally, it maintains high accuracy while reducing computational costs, making it a more efficient alternative for Bayesian decision tree models.

The main strength of this paper includes modeling prediction uncertainty using posterior distributions. The treatment of each tree structure as a unique subspace employing soft decision functions enables gradient-based sampling with HMC. This could reduce computational costs.

---

### Official Review · Reviewer_FJws · 2025-02-28
**Interesting contribution, what about computational complexity ?**

**Rating:** 7
**Confidence:** 2

**Review:**

The manuscript introduces a novel method for sampling from the posterior distribution of a Bayesian decision tree model. The method is based on a clever parametrisation of the posterior distribution, which allows to sample from the posterior distribution of the tree structure and the parameters of the model separately.

The method is compared with other methods for sampling from the posterior distribution of a Bayesian decision tree model, it is found to be among the best performing methods in terms of Mean Squared Error for real and simulated data.

As the manuscript describes a prior and a likelihood, it makes sense to evaluate it in terms of model fit. However, the main aim of the work seems to be the development of an efficient sampling method, and it would be interesting to see a comparison in terms of computational complexity of the proposed method with other methods.

The manuscript could benefit from a discussion on the selection of hyperparameters, which are quite numerous. In particular, it would be valuable to elaborate on the impact of $C_0$, which influences the number of trees included in the posterior. Additionally, insights into the depth of the trees in the posterior distribution and the proportion of possible trees explored by the sampling algorithm would be interesting.


The manuscript is generally well written and the method is clearly described. The manuscript is well structured and easy to follow. The figures are clear and informative.

---

### Official Review · Reviewer_Z7TX · 2025-03-01
**The paper presents an innovative approach to representing decision tree spaces that facilitates the use of Hamiltonian Monte Carlo for updating decision parameters. However, the paper lacks clarity in its algorithmic descriptions and does not fully address the challenge of efficiently exploring tree topologies. The performance improvements are marginal, and there is no information on computational efficiency compared to existing methods.**

**Rating:** 4
**Confidence:** 4

**Review:**

This paper introduces a novel representation of decision tree spaces that encapsulates both tree topology and decision parameters, aiming to simplify updating of the decision parameters. The proposed representation facilitates the use of Hamiltonian Monte Carlo (HMC), a gradient-based method, to sample decision parameters, building on soft decision parameters as suggested by Cochrane.
The sampling algorithm presented appears to be constructing a proposal distribution, $q$ and the difference between the posterior and the proposal are adjusted using the weights (similarly to importance sampling).

The statement, "The standard grow/prune/stay random walk method is used to propose new tree topologies to be considered, such that there is a non-zero probability of exploring any topology," could be slightly clearer. It might be more precise to state "positive probability".

I interpret this to suggest that the approach is a local proposal or a variant of the random walk methodology originally proposed by Chipman, which ensures that every state is recurrent and can be reached from any initial state. Upon reading the first paragraph on page 2, which hints at a novel representation fostering a new efficient proposal for exploring tree space, it seems the paper might suggest a breakthrough in this area. However, it appears that the critical challenge of exploring the tree space in Bayesian inference over decision trees has not been fully addressed.

In the results section, the test accuracy for DCC-Tree was only marginally better compared to HMC-DF/DFI as shown in Table 1. Additionally, while DCC-Tree demonstrated comparable performance to other methods in Table 2, Sequential Monte Carlo (SMC) seemed to excel across all datasets evaluated. This observation raises questions about the practical benefits of the proposed ideas. Without a comparison of computation times, it is unclear whether DCC-Tree offers a speed advantage over existing methods.

Minor comments:

In algorithm 1, if no tree $\mathbb{T}_m \in \mathbb{D}$ exceeds $C_0$ and $\mathbb{A} = \emptyset$, then how do we carry out step 1:  calculate utility $U_m$ for each $\mathbb{T}_m \in \mathbb{A}$ since $\mathbb{A} = \emptyset$?

On Equation (4), f and h are not clearly defined. A brief explanation would be helpful.

Similarly, $\mu_{m,k}$ and $\sigma_m$ are not clearly defined. From the context, it looks like they describe Gaussian prior over the decision parameters at the leaves. Adding a sentence to describe them would be helpful.

---

### Official Review · Reviewer_QvGx · 2025-03-01
**A worthwhile idea whose potential benefits and shortcomings are inadequately explored**

**Rating:** 5
**Confidence:** 4

**Review:**

This paper applies the divide, conquer, and combine (DCC) approach to sampling from Bayesian decision trees; they call the result DCC-Tree.  As I understand it, the intuition is simple enough: if you condition on the tree topology, it is straightforward to use HMC to efficiently sample from a relaxed set of tree parameters.  The posterior probability of a tree topology is proportional to the marginal data probability given the tree topology; this marginal probability can in principle be estimated using importance--sampling based numerical integration.  Together with a reasonable way to explore a finite set of tree topologies gives a principled way to explore the posterior on decision trees without having to mix jointly in the tree topology and parameters, which has the potential to increase sample efficiency and decrease algorithm complexity.

Some of the details of the DCC approach in general seem worth more critical examination than the paper gives them, particularly the importance sampling approach to estimating the marginal likelihood.  For example, the IS step will be more variable in topologies with more parameters, potentially biasing the posterior estimates of the form and extent of topological complexity.  But these concerns could be reasonably left aside if the paper more convincingly demonstrated the claimed benefits, so I will focus instead on the experimental results.

The paper evalutes DCC-Tree and competitive methods in terms of (a) test set accuracy and (b) sampling variability.  There are not clear gains in accuracy; even in cases where DCC-Tree is superior to other HMC methods, it is only marginally so, and it is not the best method on most of the real datsets.  The second metric, sampling variability, appears to be the papers' primary claimed contribution, since DCC-Tree is typically competitive in accuracy, but exhibits less sampling variability across MCMC runs.

I believe the authors implicitly attribute this reduced variability to improved MCMC mixing.  It may well be that DCC-Tree is mixing better than its competitors.  If this is the central claim, the authors need to show much more clearly.  Most importantly the authors do not report in any way the tradeoff between computation and accuracy across different methods.  As far as I can tell, figure 2 is the only attempt to show that the DCC-Tree is producing the same posterior estimates as RJHMC-Tree (not defined, but presumably reversible jump HMC), and is not strong evidence for the central claim of better mixing relative to the other methods.

Perhaps more fundamentally, variability from one run to the next is not necessarily the right target for Bayesian methods.  A poorly mixing chain will also not be variable from one run to the next.  More directly, some of the variability from one chain to the next will be due to real posterior variability, and some due to Monte Carlo error.  The authors make no attempt to tease out these two sources of variability.  In fact, if the authors simply want to reduce the sampling variability of the whole method for a given accuracy, the right point of comparison is with non-Bayesian tree methods.

I think a paper should make either a clear theoretical case or a clear experimental case for a method, ideally both.  I do not think the present paper quite does either.  It is very likely DCC-Tree is a better method than its competitors, but more careful evidence seems to be required.

---

### Meta-Review · Area_Chair_cTfS · 2025-03-16

**Recommendation:** Accept
**Confidence:** 4

**Metareview:**

In this paper, the authors propose a way to quantify uncertainty in decision trees. This is done through sampling both the tree topology and each local parameters. A parallel HMC is used to do so.

While most reviewers agree that this paper is interesting, it received a divided set of reviews, where most high-confidence reviews lean towards a rejection. The issues can be boiled down to evaluation and hyperparameters selection. In their rebuttal, the authors provided convincing evidence, esp. in showing the efficiency gain provided by their method.

In the final version, I urge the authors address all issues raised by the reviewers.

---

### Decision · Program_Chairs · 2025-03-18

Accept